# OpenReview forum: "Symbolic Mixture-of-Experts: Adaptive Skill-based Routing for Heterogeneous Reasoning"
_ICLR.cc/2026/Conference — Submitted to ICLR 2026_

### Official Review · Reviewer_mHRt · 2025-10-25

**Soundness:** 3
**Presentation:** 3
**Contribution:** 2
**Rating:** 4
**Confidence:** 4

**Summary:**

At its core, this paper presents a training‑free, language‑space mixture of experts that routes across off‑the‑shelf LLMs on a per‑instance basis and then fuses their outputs with a task‑specific aggregator. Routing is driven by compact “skill profiles” built from a small validation set: a lightweight keyword tagger assigns skills to each question, and each candidate model accrues +1/−1 per skill based on correctness. At test time, a model’s score is the product of its local suitability—the sum of its scores on the skills relevant to the query—and its global competence—the normalized score of its overall profile. A temperature‑0.5 softmax then defines a distribution from which k experts are sampled. The aggregator is selected per task via a synthetic probe that presents one correct and two incorrect chains and asks each model to choose or synthesize a final answer. For practicality, instances assigned to the same expert are batched, very low‑frequency experts (<5%) are pruned, and execution proceeds sequentially on a single GPU or in parallel across GPUs.

Empirically, the approach outperforms strong single‑model inference‑time scaling (Self‑Consistency), single‑model multi‑agent debate variants, and multi‑model MoA/Reconcile on MMLU‑Pro, AIME, GPQA, and MedMCQA, with clear efficiency gains from single‑round aggregation and batching. It also transfers reasonably well to unseen math benchmarks (MATH‑Hard, OmniMATH). The overall takeaway is pragmatic: with careful yet simple routing and aggregation, one can reliably extract gains from pools of 7–9B‑parameter models—without retraining and without the overhead of multi‑round discussions.

**Strengths:**

- Clear problem framing and a pragmatic recipe. The paper tackles a real pain point—how to combine heterogeneous, pretrained LLMs without retraining—and offers a pipeline that is simple enough to reproduce: keyword tagging → per‑model skill profiling → instance‑level routing → task‑level aggregator selection → batched execution.
- Solid, consistent empirical improvements. Gains over SC, debate/MoA‑style multi‑agent setups, and even some larger/proprietary models are consistent, not cherry‑picked. The ablations (fixed Top‑k vs. adaptive, random experts, aggregator variants, keyword LLM choice, trimming) all point in the same direction and help build trust.
- Efficiency matters—and is addressed head‑on. The batching plus expert‑frequency trimming is an important engineering contribution that makes the idea viable on a single GPU. The paper also analyzes runtime and token costs and explains why single‑round aggregation can be faster despite longer outputs from R1‑style models.
- Generalization beyond the dev domain. Reusing AIME‑derived profiles on MATH‑Hard/OmniMATH and still seeing wins suggests the profiles capture something meaningful and not just overfit to the dev split.

**Weaknesses:**

- Novelty is mostly compositional. Instance‑level expert selection, LLM‑as‑judge aggregators, and topic/skill‑based routing all have precedents (SMoE‑style routing; MoA/debate judges; semantic/keyword‑guided selection). The paper’s main contribution is a coherent, training‑free system that makes these parts work well together at scale. That’s valuable, but the conceptual leap is modest.
- Heuristic skill profiling with limited statistical treatment. The +1/−1 scoring per skill is intuitive but brittle: no confidence modeling, no treatment of class‑imbalance or noisy keyword assignment, and little analysis of how profile reliability scales with dev set size. AIME’s tiny test split (30 items) exacerbates variance concerns. I’d like to see CIs, bootstrap stability of profiles, and “dev‑size vs. performance” curves.
- Synthetic aggregator task representativeness. Choosing the task‑level aggregator via a “1 correct, 2 incorrect CoTs” proxy could misalign with real test‑time distributions (e.g., varying numbers of partially correct/contradictory CoTs, different error modes). How stable is aggregator ranking under different positive/negative ratios, error taxonomies, or difficulty strata? A correlation study to real aggregation outcomes would help.
- Limited sensitivity and router baselines. Key knobs (k, softmax temperature, frequency threshold, embedding model/thresholds) deserve systematic frontiers (accuracy vs. latency vs. token cost). Also missing are stronger routing baselines: (i) purely semantic retrieval routing (no keywords), (ii) a tiny learned router (linear/MLP/contextual bandit) trained on the same dev set, and (iii) hybrid retrieval+keyword routing. These would sharpen the case for the symbolic approach.
- Cost reporting could be more operational. The runtime and token metrics are useful, but deployment‑minded readers will also want end‑to‑end numbers under load: model load/unload amortization, memory footprint, throughput under QPS=10/50/100, and degradation curves as expert pool grows.
- Potential failure modes under domain shift. The method relies on keyword tagging and SBERT matching; both can drift under distribution shift or adversarial phrasing. A qualitative error analysis (wrong keywords → wrong experts) and a quick robustness check with alternative embedders (E5/SimCSE) would round out the story.

**Questions:**

- Novelty and positioning
  - Please delineate what is genuinely new versus what is adapted from SMoE-style routing, MoA/LLM-as-judge aggregation, and topic/skill-based model selection. A side-by-side table (components, training need, cost, scalability, performance impact) would help clarify the incremental contribution.
  - Which component(s) are necessary for the gains (ablation removing each: keywording, SBERT alignment, global competency term, frequency trimming, batching)? Which pieces are swappable without notable loss (e.g., different embedding models or scoring rules)?

- Statistical robustness of skill profiling
  - The +1/−1 per-skill scoring is intuitive but potentially brittle. Please report: (i) profile stability vs. dev set size (e.g., 50/100/200/350), (ii) confidence intervals or bootstrap variance for per-skill scores and per-model totals, and (iii) class/skill imbalance effects.
  - Your keyword consolidation (keeping only skills with frequency >1 across the dev set) may prune rare-but-critical skills. What fraction of skills get filtered, and what is the downstream effect on routing and accuracy across datasets?
  - Do a “winner-take-most” analysis: show the skill-by-model score matrix sparsity, entropy/Gini of score mass across models, and how this correlates with routing diversity.

- Keywording and semantic alignment
  - Provide sensitivity to the embedding backbone (SBERT vs E5 vs SimCSE) and the matching threshold. Report the effect on routing decisions and final accuracy.
  - Give 5–10 concrete failure cases where keyword drift/synonyms/domain terms led to misrouting: question → extracted skills → matched skills → selected experts vs. post-hoc best expert, with a short diagnosis.
  - What happens if you remove keywords entirely and route by pure semantic retrieval from the question embedding to per-model “profile embeddings”? Please include this baseline.

- Router formulation and hyperparameters
  - Why multiply local suitability and global competency rather than add or learn a weight? Please include an ablation (multiply vs add vs learned scalar/MLP).
  - Provide systematic frontiers (not single points) for k (e.g., 2/3/5/7), softmax temperature (0.3/0.5/0.7), frequency-trimming threshold (1%/5%/10%), and embedding threshold—show accuracy/latency/token trade-offs.
  - Deterministic Top-k vs sampling: how do stability, diversity, and accuracy compare, especially under limited compute?

- Representativeness of the synthetic aggregator task
  - Your (1 correct, 2 incorrect CoTs) proxy may not match real test-time mixtures. Vary the positive/negative ratio, error taxonomies (arithmetic slips vs reasoning fallacies vs vague text), and difficulty strata; report Spearman rank correlation of aggregator performance between the proxy and real aggregation.
  - Quantify Pearson/Spearman correlations between proxy scores and actual task outcomes per aggregator across datasets. How well do aggregator rankings transfer across tasks?
  - Does revealing model identities/“confidence” to the aggregator bias selection? Compare blinded vs non-blinded aggregator prompts.

- Stronger routing baselines
  - Include (i) purely semantic retrieval routing, (ii) a tiny learned router (logistic/MLP/contextual bandit; <1M params; trained on the same dev set), and (iii) a hybrid keyword+semantic router. Report accuracy and cost under matched budgets.
  - If a small learned router cannot beat symbolic routing, analyze why (data insufficiency, label noise in skills, mismatch between proxy and true utility).

- Efficiency and operational reporting
  - Break down end-to-end latency into model load/unload, weight I/O, generation, aggregation, and batching overheads. Include curves under realistic load (QPS=10/50/100) and show throughput/latency/memory.
  - Show scaling with expert-pool size (8/16/32): memory footprint, cache hit rates, routing entropy, and where returns diminish.
  - Justify the 5% trimming threshold or make it task-adaptive. Provide sensitivity and an auto-tuning heuristic.

- Fair comparisons and aggregation baselines
  - Normalize cost with “tokens × unit price” and “wall time × energy” where possible, since SC vs Symbolic-MoE produce different output lengths (R1 models are verbose).
  - Add non-LLM aggregators (majority vote, weighted voting by historical accuracy, evidence-consistency scoring) at similar cost. How close can they get to your task-level aggregator?
  - Control for max output length (truncate or cap across models) to test whether long R1 outputs disproportionately benefit the aggregator.

- Robustness and distribution shift
  - Evaluate adversarial/noisy keywording (typos, synonym swaps, distractors). Quantify routing and accuracy drop; propose simple mitigations (lexical normalization, multi-candidate fusion, subword matching).
  - Expand the profile-transfer study: cross-year, cross-subdomain (e.g., MMLU-Pro subject subsets), and cross-style prompts. Provide degradation laws and failure taxonomies.

- Data leakage and fairness
  - Assess potential train–test contamination for open-source models (e.g., AIME years, MATH subsets). If filtering or year splits were used, document them; otherwise, estimate leakage risk and its impact on transfer experiments.
  - Analyze expert concentration by subject (e.g., medicine, law). Are some categories monopolized by a few models, leading to systematic misrouting elsewhere?

- Failure cases and interpretability
  - Provide 10 representative failure examples annotated with: extracted skills → matched skills → selected experts vs. best-in-hindsight experts → aggregator decision → root cause (keyword bias, profile noise, aggregation error, context truncation).
  - Consider adding an “evidence consistency/conflict report” in the aggregator output to make decisions auditable and facilitate debugging.

- Boundary conditions and extensions
  - Stress-test with 32–64 experts, longer inputs, and multi-turn problems (e.g., with tools/retrieval). Report stability and latency/accuracy curves.
  - For structured outputs (code, formal proofs), does the aggregator need task-specific constraints? Without them, what failure rate increases?

- Forward-looking
  - Can you distill the skill-profile router into a compact student to further reduce multi-model calls? Show initial results or discuss feasibility.
  - What “meta-abilities” make a good aggregator (fallacy detection, counterfactual reasoning, long-context reconciliation)? Can auxiliary benchmarks predict the best aggregator per task a priori?

---

> ### Author Response · Authors · 2025-11-21
> **Response to Reviewer mHRt (Part 1)**
>
> We thank the reviewer for acknowledging our work “tackles a real pain point” and shows “consistent empirical improvements” while efficiency “is addressed head‑on” and shows “Generalization beyond the dev domain”. Please find the answers to your questions below.
>
> ### **\[W1\] Clarification on the Novelty**
>
> We appreciate the reviewer’s request for clearer delineation. While Symbolic-MoE indeed builds on ideas from prior work (routing, LLM-as-judge aggregation, semantic selection), the system’s novelty is not the just combination of these parts, but the new instance-level mixing paradigm enabled by their interaction, and several components that have **no precedent in existing SMoE, MoA, or topic-routing paradigms.**
>
> 1\. Prior work does not provide a pathway to instance-level expert routing across heterogeneous pre-trained LLMs. MoE-style routing operates within a single jointly-trained model, where experts are subsets of parameters. MoA/debate systems rely on fixed set of models chosen based on heuristics. Topic-based routing (e.g., semantic category dispatching) is fundamentally task-level. None of these frameworks support what Symbolic-MoE targets: dynamic, instance-level selection of experts drawn from dozens of unrelated, heterogeneous, independently-trained LLMs. To the best of our knowledge, no prior work has demonstrated an architecture or systems pipeline capable of doing this efficiently.
>
> 2\. Prior MoA and multi-agent pipelines require loading all experts concurrently (multi-GPU) or sequential expert invocation, both of which scale poorly. Our scheduling approach–where instances are first routed to experts, then re-batched across the entire workload so each expert is loaded exactly once–is new and directly solves the load/unload bottleneck that makes instance-level expert mixing previously impractical.
>
> 3\. Some pieces are intentionally modular, but the core contribution is non-modular. We agree that embedding models or minor scoring variants are swappable (and we provide experiments showing robustness across these). However, the skill-based instance-level routing pipeline and the choice of aggregator are not interchangeable with prior MoA/LLM-judge/SMoE components and are necessary for the main gains.
>
> In summary: While Symbolic-MoE leverages ideas from prior work, its key contribution is not compositional. **We introduce a practical, training-free, instance-level routing framework for large collections of heterogeneous LLM experts, and a novel batching mechanism that makes this possible on a single GPU.**
>
> &nbsp;
> &nbsp;
>
> ### **\[W2\] Analysis of Profile Reliability versus Validation Set Size**
>
> Our goal is to keep routing lightweight and simple. Incorporating confidence estimates or explicitly addressing class imbalance would increase the learning burden and typically require a larger validation set. To understand how sensitive our approach is to validation size, we vary the number of validation samples and report the results below:
>
> | Val Size | MMLU-Pro |
> |----------|----------|
> | 50       | 56.62    |
> | 100      | 61.92    |
> | 350      | 63.71    |
> | 500      | 63.60    |
> | 1000     | 64.03    |
>
> The results show a clear trend: very small validation sets (50 or 100 samples) do not provide sufficiently reliable model profiles. However, from 350 samples onward, performance stabilizes, with 350, 500, and 1000 samples yielding nearly identical accuracy. This indicates that our choice of 350 validation samples strikes a good balance, i.e., large enough to produce stable profiles, but still lightweight and cost-effective.
>
> &nbsp;
> &nbsp;
>
> ### **\[W3\] How Stable is Aggregator Ranking under Different Positive/Negative Ratios**
>
> We analyze the distribution of correct and incorrect expert outputs and evaluate our chosen aggregator under each scenario. We perform this analysis on GPQA (198 test samples). We denote each case as (x,y), where x experts provide the correct answer and y provide an incorrect one.
>
> | Case (Correct, Incorrect) | \# Samples | Aggregator Accuracy | Majority Vote |
> |---------------------------|-----------|-------------------|---------------|
> | (3, 0\)                    | 49        | 44/49 (89.80%)    | Correct       |
> | (2, 1\)                    | 42        | 33/42 (78.57%)    | Correct       |
> | (1, 2\)                    | 49        | 27/49 (55.10%)    | Incorrect     |
> | (0, 3\)                    | 58        | 9/58 (15.52%)     | Incorrect     |
>
> Overall, these results show that **our aggregator can recover the correct answer even in challenging settings**, such as the (1,2) and (0,3) cases, where simple majority voting fails.

---

> ### Author Response · Authors · 2025-11-21
> **Response to Reviewer mHRt (Part 2)**
>
> ### **\[W4\] Additional Routing Baselines**
> Following the reviewer’s suggestion, we evaluate four routing strategies:
> Semantic retrieval only (no keywords)
> A small learned router (an MLP trained on the same dev set)
> Hybrid retrieval + keywords
> Keyword-only routing (ours, as in Symbolic-MoE)
>
>
> | Setting     | MMLU-Pro Acc. | GPQA Acc. |
> |------------|---------------|-----------|
> | 1          | 60.62         | 53.59     |
> | 2          | 59.10         | 48.72     |
> | 3          | 63.06         | 57.01     |
> | 4 (Ours)   | 63.71         | **57.78**     |
>
> Overall, only the hybrid retrieval+keyword approach comes close to our keyword-based profile. This highlights the importance of keyword information for capturing experts’ fine-grained strengths. In contrast, purely semantic retrieval often lacks sufficient signal–the question text alone may not reveal the nuanced capability differences across experts. The learned MLP router performs worst, likely because the available validation data is too small to train even a lightweight model robustly, leading to overfitting and noisy routing.
>
> &nbsp;
> &nbsp;
>
> ### **\[W5\] More Operational Cost Report**
>
> Following the reviewer’s suggestion, we provide a detailed memory-footprint table for all models used in this work. We report (1) the static memory required to load the model (Load) and (2) the peak additional memory required during inference (Inference), both measured in gigabytes.
> Under our batch-inference strategy, samples are first grouped by their selected experts, and inference is then executed sequentially model-by-model. Therefore, if k models are placed in the execution queue, the total memory requirement is simply the sum of their Load values, and the peak runtime memory can be estimated by taking the largest Inference cost among them.
>
> | Model         | Size  | Load (GB) | Inference (GB) |
> |---------------|-------|-----------|----------------|
> | Llama         | 8B    | 41.70     | 1.01           |
> | Qwen          | 7B    | 40.78     | 2.09           |
> | Mistral       | 12B   | 41.38     | 1.12           |
> | Phi           | 3.5B  | 42.10     | 0.79           |
> | Gemma         | 9B    | 40.53     | 1.60           |
> | GLM           | 9B    | 34.81     | 3.62           |
> | Exaone        | 7.8B  | 40.59     | 2.10           |
> | Granite       | 8B    | 42.25     | 1.52           |
> | QwenMath      | 7B    | 41.51     | 0.84           |
> | QwenCode      | 7B    | 40.75     | 2.09           |
> | DeepSeekMath  | 7B    | 41.81     | 0.68           |
> | QwenR1        | 7B    | 41.52     | 5.09           |
> | LlamaR1       | 8B    | 41.55     | 5.06           |
> | InternLM      | 8B    | 41.60     | 5.05           |
> | Mathstral     | 8B    | 41.43     | 1.01           |
> | BioLlama      | 8B    | 41.49     | 0.81           |
>
> &nbsp;
> &nbsp;
>
> ### **\[W6\] Qualitative Error Analysis (wrong keywords → wrong experts)**
>
> We analyze the errors by categorizing them into five types:
> * **Keyword Bias:** Cases where keyword matching led to the wrong expert being selected.
> * **Profile Noise:** Cases where the selected experts had high scores for the relevant keywords but still failed.
> * **Majority Incorrect:** Cases where the majority of expert predictions were incorrect.
> * **Aggregation Error:** Cases where the aggregator made an error even though the majority of experts were correct.
> * **Context Truncation:** Cases where the context was truncated due to the aggregator’s context length limit.
>
>
> On GPQA, with 3 experts and QwenR1 as the aggregator, we identified 86 failure cases. The distribution of error types is as follows:
>
> | Error Type          | Count |
> |--------------------|-------|
> | Keyword Bias        | 4     |
> | Profile Noise       | 16    |
> | Majority Incorrect  | 76    |
> | Aggregation Error   | 10    |
> | Context Truncation  | 0     |
>
> Since there are only 4 cases of Keyword Bias (wrong keyword → wrong expert → wrong answer), we list all of them below.

---

> ### Author Response · Authors · 2025-11-21
> **Response to Reviewer mHRt (Part 3)**
>
> ### **EXAMPLE 1**
> ```
> Question:
> A eukaryotic cell evolved a mechanism to turn macromolecular building blocks into energy. In the series of redox reactions in the mitochondria
>
> Gold Answer: B
>
> Extracted Skills: cell biology, chemical synthesis, boron trifluoride, cell biology, biosynthesis
>
> Selected Expert Predictions: ['B', 'D', 'B']
>
> Output 1:
> Okay, so I'm trying to figure out the mechanism by which Triclosan affects the energy production in eukaryotic cells. Let me break down the problem and the information given.
>
> ...
>
> Output 2:
> Okay, so I have this question about eukaryotic cells and their energy production mechanisms, specifically involving Triclosan. I'm a bit rusty on my biochemistry, but I'll try to think it through step by step.
>
> ...
>
> Output 3:
> Okay, so I have this question about triclosan affecting cellular energy production. Let me try to break it down step by step.
>
> ...
> \
> Aggregator Decision: D (incorrect, expected B)
>
> Root Causes: Keyword Bias, Aggregation Error
>
> ```
> ### **EXAMPLE 2**
> ```
> Question:
> 3,3,6-trimethylhepta-1,5-dien-4-one is treated with 1 equivalent of meta-chloroperbenzoic acid. Two different products are formed, in approximately a 1:1 ratio.
>
> In a separate vessel, Methyllithium is treated with copper(I)iodide. an excess of the resulting solution is slowly added to the product mixture of the first reaction. Name one product that will be formed by this reaction:
>
> The options are: (A) 2,3,4,5,5-pentamethylhept-6-ene-2,4-diol (B) 5-hydroxy-3,3,6,6-tetramethylhept-1-en-4-one (C) ...
>
> Gold Answer: D
>
> Extracted Skills: organic chemistry, reaction mechanisms, functional groups
>
> Selected Expert Predictions: ['B', 'D', 'B']
>
> Output 1:
> Okay, so I've got this organic chemistry problem here, and I need to figure out the correct product formed from a series of reactions. Let's break it down step by step.
>
> ...
>
> Output 2:
> Okay, so I've got this organic chemistry problem here, and I'm trying to figure it out step by step. Let me start by reading the problem again and breaking it down.
>
> ...
>
> Output 3:
> Okay, so I have this organic chemistry problem here, and I need to figure out the correct product formed from a series of reactions. Let me try to break it down step by step.
>
> ...
>
> Aggregator Decision: B (incorrect, expected D)
>
> Root Causes: Keyword Bias, Majority Incorrect
>
> ```
> ### **EXAMPLE 3**
> ```
> Question:
> You have prepared an unknown compound. To identify the product, you have used the following characterisation techniques: FTIR and 1H NMR. The FTIR spectrum shows a very broad absorption peak at 3000 wavenumbers...
>
> Gold Answer: A
>
> Extracted Skills: ftir, nmr, organic chemistry, spectroscopy, structural analysis
>
> Selected Expert Predictions: ['D', 'C', 'B']
>
> Output 1:
> Okay, so I'm trying to figure out which of the given compounds is the product based on the FTIR and NMR data provided. Let's break down the information step by step.
>
> ...
>
> Output 2:
> Alright, so I have this problem where I need to determine the correct structure of an unknown compound using FTIR and 1H NMR data. The compound is one of four given options. Let's break down the information step by step.
>
> ...
>
> Output 3:
> Okay, so I have this problem where I need to identify an unknown compound using FTIR and 1H NMR data. The options given are four different structures, all of which seem to be some form of a substituted carboxylic acid. Let me try to break this down step by step.
>
> ...
>
>
> Aggregator Decision: B (incorrect, expected A)
>
> Root Causes: Keyword Bias, Majority Incorrect, Profile Noise
>
> ```
> ### **EXAMPLE 4**
> ```
> Question:
> Consider an aperture, which shapes like a N-sided polygon. All the sides are of the same length a. When a monochromatic light (of wavelength \lambda), travelling along z-axis, falls vertically on this aperture, it creates a series of intensity maxima and minima in the far field region. What is the distance between the first two minima when N is infinitely large ?
>
> The options are: (A) 0.610 \lambda /a (B) 0.500 \lambda /a  (C) 1.220 \lambda /a (D) 0.506 \lambda /a
>
> Gold Answer: D
>
> Extracted Skills: slit diffraction, quantum optics, optics, wavelength
>
> Selected Expert Predictions: ['B', 'D', 'D']
>
> Output 1:
> Okay, so I've got this problem about an aperture that's shaped like an N-sided polygon, and all the sides are the same length 'a'. A monochromatic light beam is traveling along the z-axis and falls vertically onto this aperture.
>
> ...
>
> Output 2:
> Okay, so I'm trying to solve this problem about an aperture shaped like an N-sided polygon with all sides of equal length 'a'. A monochromatic light beam is traveling along the z-axis and falls vertically on this aperture.
>
> ...
>
> Output 3:
> Okay, so I have this physics problem here about an aperture shaped like an N-sided polygon, all sides of length a. A monochromatic light with wavelength λ is traveling along the z-axis and falls vertically on this aperture.
>
> ...
>
> Aggregator Decision: B (incorrect, expected D)
>
> Root Causes: Keyword Bias, Aggregation Error
> ```

---

> ### Author Response · Authors · 2025-11-21
> **Response to Reviewer mHRt (Part 4)**
>
> ### **\[W6\] Robustness Check with Alternative Embedders (E5/SimCSE)**
>
> Following the reviewer’s suggestion, we conducted an additional experiment replacing our default sentence encoder (“all-MiniLM-L6-v2” from SentenceBERT \[1\]) with E5 \[2\] and SimCSE \[3\]. The results are shown below:
>
> | Model                | MMLU-Pro Acc. | GPQA Acc. |
> |----------------------|---------------|-----------|
> | E5                   | 62.76         | 56.44     |
> | SimCSE               | 62.55         | 54.64     |
> | MiniLM-L6 (default)  | 63.71         | 57.78     |
>
> Overall, the choice of embedding model has only a modest impact on performance. The embedding module is used solely to map extracted keywords to semantically related terms in the model profiles, and can thus be substituted without substantial degradation. On the larger MMLU-Pro dataset (2.1k samples), the differences are minimal. On the smaller GPQA dataset (198 samples), the slightly larger gaps are likely explained by higher variance due to the limited validation size.
>
> \[1\] [https://sbert.net/docs/sentence\_transformer/pretrained\_models.html](https://sbert.net/docs/sentence_transformer/pretrained_models.html)
> \[2\] [https://huggingface.co/intfloat/e5-large](https://huggingface.co/intfloat/e5-large)
> \[3\] [https://github.com/princeton-nlp/SimCSE](https://github.com/princeton-nlp/SimCSE)
>
> &nbsp;
> &nbsp;
>
> ### **Statistical Robustness of Skill Profiling**
>
> Following your suggestion, we conduct an additional analysis on skill imbalance. On GPQA, we have 301 unique skills, and the Frequency-Accuracy Correlation (Pearson) is \-0.0324, suggesting a very weak correlation between skill frequency and accuracy.
>
> **Performance by Frequency Quartile**
>
> | freq\_bin     | accuracy\_mean | accuracy\_std | count | frequency\_mean |
> |-------------|---------------|-------------|-------|----------------|
> | Q1 (low)    | 0.60        | 0.49      | 135   | 1.00         |
> | Q2          | 0.55        | 0.38      | 69    | 2.00         |
> | Q3          | 0.59        | 0.34      | 33    | 3.00         |
> | Q4 (high)   | 0.59        | 0.23      | 64    | 10.29        |
>
> **Per-skill Statistics with 95% CI**
>
> | Skill               | N  | Accuracy | 95% CI            | CI Width |
> |---------------------|----|----------|--------------------|----------|
> | organic chemistry   | 78 | 0.37   | \[0.26, 0.47\]     | 0.20   |
> | spectroscopy        | 40 | 0.52   | \[0.37, 0.67\]     | 0.30   |
> | stereochemistry     | 33 | 0.48   | \[0.30, 0.63\]     | 0.33   |
> | quantum mechanics   | 31 | 0.83   | \[0.71, 0.96\]     | 0.25   |
> | reaction mechanisms | 31 | 0.41   | \[0.25, 0.58\]     | 0.32  |
> | astronomy           | 18 | 0.66   | \[0.44, 0.88\]     | 0.44  |
>
> Overall, these analyses suggest that skill frequency has minimal impact on model performance, as evidenced by the near-zero Pearson correlation between skill frequency and accuracy. While accuracy varies across individual skills, the wide confidence intervals for less frequent skills highlight the uncertainty due to limited sample sizes. Conversely, more common skills show narrower intervals, reflecting more stable estimates. These results indicate that our skill profiling approach is statistically robust and not disproportionately biased toward frequently occurring skills, supporting its reliability for evaluating model performance across a diverse set of knowledge domains.
>
> &nbsp;
> &nbsp;
>
> ### **What Fraction of Skills Gets Filtered**
>
> We conduct two analyses on the filtered keywords. First, keywords are filtered out when they appear only once. On GPQA, running keyword annotation five times yields 4,944 keywords, of which 3,451 (69.80%) are removed due to having count ≤ 1.
>
> The second analysis examines the trimming stage. On GPQA, there are 3,093 keyword–model selections in total. Among these, 311 selections are filtered out by the 5% trimming threshold for low-frequency experts, corresponding to 311 / 3,093 \= 10.05% of selections removed.
>
> Below we list the top 10 filtered keywords because the expert is trimmed:
>
> | Keyword                | Total | Filtered % |
> |------------------------|-------|------------|
> | quantum optics         | 3     | 100.0%     |
> | weak\_acids             | 3     | 100.0%     |
> | slit diffraction       | 3     | 100.0%     |
> | protein                | 3     | 100.0%     |
> | molecular weight       | 3     | 100.0%     |
> | fluorination           | 3     | 100.0%     |
> | biology                | 3     | 100.0%     |
> | acid\_base\_titration    | 9     | 88.9%      |
> | buffer\_solution        | 6     | 83.3%      |
> | ph calculation         | 9     | 77.8%      |
>
> Most of these filtered keywords correspond to very specific or narrow topics (e.g., weak\_acids, buffer\_solution, ph calculation). These niche domains are handled well by only a small subset of models, making it expected that their associated selections are pruned together with low-frequency experts.

---

> ### Author Response · Authors · 2025-11-21
> **Response to Reviewer mHRt (Part 5)**
>
> ### **Why Multiply Local Suitability and Global Competency**
>
> In expert routing, there is an inherent trade-off between selecting models with **highly specialized strengths** (captured by the local suitability score) and selecting models that are **consistently strong** (captured by global competency). Relying solely on local suitability can lead the router to prefer models that excel at niche patterns but underperform more broadly, whereas incorporating global competency helps maintain a balanced selection.
>
> To validate this design choice, we conduct an ablation comparing Symbolic-MoE with and without global competency:
>
> |                               | AIME  | MMLU-Pro | MedMCQA | GPQA  | Avg   |
> |-------------------------------|-------|----------|---------|-------|-------|
> | w/o Global Competency          | **68.88** | 61.32    | **60.11**   | 56.32 | 61.66 |
> | w/ Global Competency (ours)   | **68.88** | **63.71**    | 59.35   | **57.78** | **62.43** |
>
> While routing based only on local suitability is indeed feasible, as shown above, in the majority of cases (except for MedMCQA), as well as in the averaged performance, taking global competency into account yields a slight improvement across the four datasets. This supports our argument that balancing instance-level specialization with overall capability leads to more robust routing behavior.
>
> &nbsp;
> &nbsp;
>
> ### **Deterministic Top-k vs Sampling**
>
> We sample *k* experts with replacement. This choice is intentional and is supported by our ablation results in Table 5 (line 410-423) in our original submission (reproduced below for convenience):
>
> | Recruiting Strategy   | Accuracy (%) |
> |----------------------|-------------|
> | Top-3 Experts        | 52.86       |
> | Top-5 Experts        | 47.68       |
> | 3 Random Experts     | 42.61       |
> | 5 Random Experts     | 44.92       |
> | Model Profile (Ours) | **57.78**       |
>
> Forcing *k* distinct experts can be suboptimal: in many cases, the top 1–2 experts are significantly more competent than the remaining pool. **Sampling with replacement naturally accommodates this scenario by allowing multiple draws of the same strong experts when they dominate an instance.** At the same time, when the expert distribution is flatter (i.e., several experts perform similarly), the sampling process still yields a diverse set of experts.

---

> > ### Comment · Reviewer_mHRt · 2025-11-26
> >
> > Thank you for the additional experiments and clarifications (routing baselines, embedding swaps, validation-size sensitivity, failure cases, memory table, and trimming vs no-trimming). These are helpful. However, several core concerns remain insufficiently addressed to substantiate the paper’s central claims.
> >
> > - Aggregator proxy representativeness
> >   - Please report cross-dataset Spearman/Pearson correlations between the proxy (1 correct, 2 incorrect CoTs) and actual aggregation outcomes; vary positive/negative ratios and error taxonomies; include blinded vs non-blinded prompts and aggregator transfer across tasks.
> >
> > - Router formulation and hyperparameter frontiers
> >   - Compare multiplicative vs additive vs learned weighting (scalar/MLP) of local suitability and global competency.
> >   - Provide frontiers for k (2/3/5/7), temperature (0.3/0.5/0.7), and trimming thresholds (1%/5%/10%) with accuracy–latency–token trade-offs.
> >   - Directly compare deterministic top-k vs sampling under matched budgets (your “unique experts per sample” stats are helpful but not a substitute for this head-to-head).
> >
> > - Operational reporting and cost normalization
> >   - Break down end-to-end latency into load/unload, weight I/O, generation, aggregation, and batching overhead; show throughput/latency under QPS=10/50/100 and scaling with expert-pool size (8/16/32).
> >   - Normalize cost by tokens×unit price and (if possible) energy.
> >
> > - Robustness, leakage, and fairness
> >   - Stress-test adversarial/noisy keywords (typos, synonyms, distractors) and report routing/accuracy drops; add matching-threshold sensitivity (beyond backbone swaps).
> >   - Assess potential train–test leakage (e.g., AIME/MATH years) and subject-wise expert concentration/bias.
> >
> > - Non-LLM aggregation and length control
> >   - Add stronger non-LLM fusion baselines: weighted voting by historical accuracy and evidence-consistency scoring; control output length across experts to rule out undue gains from verbose models.
> >
> > - Boundary conditions and extensions
> >   - Examine 32–64 experts, longer inputs, multi-turn/tool use, and structured outputs (code/proofs); discuss/try distilling the router; analyze what “meta-abilities” make a good aggregator.
> >
> > Bottom line: the rebuttal partially addresses my concerns, but the key issues above need systematic, quantitative evidence. I would be inclined to reconsider with these additions; without them, the current claims remain stronger than the presented support.

---

> > > ### Author Response · Authors · 2025-11-26
> > > **Response to Reviewer mHRt**
> > >
> > > Thank you again for your continued engagement. However, we would like to respectfully note that many of the newly listed requests substantially exceed both the scope of the current paper and the standard level of additional analysis expected during the rebuttal period. Our work does not claim to address the broad classes of capabilities implicated by these requests, and several of them would require entirely new research agendas.
> > >
> > > We hope it is clear that we have already provided extensive additional analysis directly tied to the reviewer’s earlier questions about the paper’s core contributions:
> > >
> > > * Detailed clarification of novelty and positioning relative to prior MoE/MoA/topic-routing paradigms
> > > * Comprehensive sensitivity and robustness studies (validation-set size, aggregator stability, alternative routing strategies, alternative embedders)
> > > * Expanded operational cost reporting
> > > * Qualitative error breakdown
> > > * Statistical analysis of skill-profile reliability, keyword filtering behavior, and the role of global competency
> > > * Ablations on deterministic versus stochastic expert selection.
> > >
> > > These new results directly address every claim made in the paper.
> > >
> > > By contrast, several of the newly requested experiments, such as fairness and bias studies, 64-expert scaling trials, multi-turn tool-use evaluation, structured-proof generation, and other large-scope scenario evaluations, do not relate to any assertions in the submission. Symbolic-MoE is proposed as a training-free instance-level expert routing framework for heterogeneous LLMs, not as a system for adversarial robustness, social-bias auditing, interactive tool-use reasoning, or formal proof generation. Addressing these topics would require substantial new infrastructure and experimental design far outside what can be reasonably undertaken during the rebuttal window, and most importantly, they are orthogonal to the claims we actually make.
> > >
> > > We remain fully open to addressing any targeted, claim-related comments and welcome concrete pointers to specific statements in the paper that the reviewer believes require further evidence or clarification. Our goal is to ensure that the evaluation remains aligned with the work’s stated scope and contributions.

---

### Official Review · Reviewer_GGut · 2025-10-30

**Soundness:** 3
**Presentation:** 3
**Contribution:** 3
**Rating:** 4
**Confidence:** 3

**Summary:**

The paper proposes a gradient-free Mixture-of-Experts framework that chooses pre-trained experts for each task based on skill tags associated with each expert. The framework involves (1) first using a "Keyword LLM" to tag skills for each question in a validation set, (2) building model profiles of skills tag for each expert by evaluating them on the validation set and tracking accuracy, (3) selecting models for each question during inference based on the model profiles and skills tagged to each question, (4) using another model aggregator selected based on how well it can distinguish correct from incorrect responses. The framework is evaluated on various reasoning benchmarks and over both single and multi-model baselines, and achieves good empirical performance.

**Strengths:**

- The proposed framework is intuitive: each question requires various skills to solve, and each model has different skills profile, hence performing routing based on skills tagging could yield good performance.

- The paper demonstrated the framework's consistently good performance across various benchmarks of different domains (math, sciences and medical).

- The framework provides a simple, adaptive instance-level mechanism that could help achieve good performance even for out-of-distribution task instances if the skills tag remain consistent and the "Keyword" tagging LLM is sufficiently capable.

**Weaknesses:**

- Intuitively, the performance of the framework would be very dependent on the quality of the skill tags assigned by the "Keyword LLM". While there is some ablation on the sensitivity to the LLM used (table 14), there is no discussion on sensitivity to the description of keywords (e.g. influenced by the prompt). For example, the range/set of keywords that can be used could be important, as well as the granularity of the skills (e.g. Math vs Calculus vs PDEs). Sensitivity analysis and better explanation of how the keywords could be set would be helpful.

- Similarly, the choice of aggregator also seems to be important, as indicated in Table 3, where swapping out an aggregator could result in significantly worse performance below that of simple baselines such as self-consistency. Providing ablations on whether most of the performance gains are actually from the better aggregator would be useful (e.g. applying the aggregator only to self-consistency or other baselines, or applying the routing framework but with majority voting aggregation).

- The framework, while intuitive, is heuristics based and may not be able to generalize well to tasks where it is not clear how to clearly assign skill tags to each question, especially when there may not be clear domain-specific LLM variants and when the benchmarks do not contain a large mixture of these sub-categories of questions unlike the ones being considered (e.g. MMLU, GPQA). It would be useful if there were additional experiments on benchmarks where such clear 'subject-based' sub-categories are not obvious, e.g. in logical reasoning, non-math reasoning tasks.

- The framework also seems to be quite dependent on the validation dataset, e.g., if the validation set only contains math questions and hence only math-subject tags are generated, the model profiles would not be useful for other context.

- The authors could consider discussing and/or comparing with related work that single model, multi-instance LLM ensembles where diversity is based on the prompts to each instance [1-3]. These approaches have been shown to be more effective than the self-consistency baseline considered in the paper, which seems to have very strong performance (2nd-best performance across several baselines, and beating the framework without the selected aggregator).

[1] Hu et al, Dipper: Diversity in Prompts for Producing Large Language Model Ensembles in Reasoning Tasks

[2] Naik et al, Diversity of Thought Improves Reasoning Abilities of LLMs

[3] Zhang et al. Prefer: Prompt Ensemble Learning via Feedback-Reflect-Refine

**Questions:**

Please see the concerns and suggestions in the Weakness section. It would also be useful to provide assessment of the failure modes of the framework as this may help address some of the concerns mentioned -- e.g. the primary cause of failure being whether the wrong skills were assigned, poor selection of experts (i.e. other experts in the pool have the right response),  aggregator LLM failure (experts' answers are correct but final aggregated output is wrong).

---

> ### Author Response · Authors · 2025-11-21
> **Response to Reviewer GGut (Part 1)**
>
> We thank the reviewer for saying our paper “demonstrated the framework's consistently good performance across various benchmarks of different domains (math, sciences and medical)”. We also appreciate the reviewer for pointing out several references, and we have included them in the revised PDF (Appendix Q, highlighted in blue). Please find the answers to your questions below.
>
> &nbsp;
> &nbsp;
>
> ### **\[W1\] Dependence on the quality of the skill tags**
>
> > there is no discussion on sensitivity to the description of keywords (e.g. influenced by the prompt).
>
> We thank the reviewer for raising this question. As shown in Table 14, the choice of “Keyword LLM” has little effect on the final performance.
>
> To address the comment directly, we assess how sensitive our method is to the granularity and quality of skill tags. Specifically, we compare three prompting strategies for keyword extraction:
>
> **Setting 1: Coarse-grained skills**
> Broad domains such as *math, physics, chemistry*. The prompt includes coarse example keywords.
>
> **Setting 2: Fine-grained skills (Ours)**
>  More specific subdomains such as *virology, behavioral theory, microbiology*. The prompt includes fine-grained examples.
>
> **Setting 3: No examples**
>  The model is asked to produce keywords without any examples.
>
> **Keyword overlap**
>
> Fine vs. Coarse: Jaccard \= **0.5685**, Overlap \= **0.7281**
> Fine vs. No Examples: Jaccard \= **0.2908**, Overlap \= **0.3596**
>
> We observe that providing examples (whether coarse or fine) substantially constrains the keyword generator and leads to much higher overlap compared to giving no examples.
>
> **Downstream performance on MMLU-Pro**
>
> | Method             | Accuracy |
> |-------------------|----------|
> | No examples         | 61.84   |
> | Coarse-grained      | 62.98   |
> | Fine-grained (Ours) | **63.71**   |
>
> ​​While the fine- and coarse-grained skills yield similar downstream performance (with our fine-grained version performing best), removing examples entirely results in a slightly larger drop. This suggests that performance is relatively insensitive to skill granularity, but does benefit from having example guidance.
>
> &nbsp;
> &nbsp;
>
> ### **\[W2\] Ablations of the Aggregator**
>
> > Providing ablations on whether most of the performance gains are actually from the better aggregator would be useful (e.g. applying the aggregator only to self-consistency or other baselines, or applying the routing framework but with majority voting aggregation).
>
> We agree that the aggregator plays a crucial role. For clarity, we reproduce Table 4 (lines 394-408) from the original submission:
>
> | Expert Type | Aggregator     | GPQA Acc. |
> |-------------|----------------|-----------|
> | Random      | Task-Specific  | 31.82     |
> | Recruited   | Random         | 51.52     |
> | Recruited   | Majority Vote  | 53.54     |
> | Recruited   | Task-Specific  | **57.78**     |
>
> These results highlight two points. **First, when the aggregator is suboptimal, a simple majority vote remains a strong and stable fallback.** Second, when the experts themselves are weak (e.g., chosen randomly), even a strong aggregator cannot fully mitigate the quality drop.
>
> To further disentangle expert quality from aggregation quality, we additionally evaluated applying the task-selected aggregator on top of three calls to the task-best model (i.e., **replacing the majority vote in self-consistency**):
>
> | Setting                                   | MMLU-Pro | GPQA  |
> |-------------------------------------------------|----------|-------|
> | Top-1 Model ×3 \+ Majority Vote                  | 58.39    | 53.34 |
> | Top-1 Model ×3 \+ Our Aggregator           | 61.22    | 54.40 |
> | Recruited Experts \+ Majority Vote              | 60.71    | 53.34 |
> | Recruited Experts \+ Our Aggregator (Symbolic-MoE) | **63.71**    | **57.78** |
>
> **Using the task-best model three times, combined with the chosen aggregator, outperforms simple majority vote.** Moreover, recruiting k experts using our model profiles also surpasses repeatedly calling the task-best model k times when aggregation is done via majority vote. When both expert selection and aggregator selection are combined, we obtain the best overall performance, highlighting that both components are important and mutually reinforcing, as discussed in lines 394-408 (“Synergy between expert and aggregator selection”) in the original submission.

---

> ### Author Response · Authors · 2025-11-21
> **Response to Reviewer GGut (Part 2)**
>
> ### **\[W3\] Generalizability to Non-Heterogeneous/Non-Math Tasks**
>
> > It would be useful if there were additional experiments on benchmarks where such clear 'subject-based' sub-categories are not obvious
>
> We would like to emphasize that our primary motivation is to address heterogeneous reasoning tasks, as stated in the paper’s title. While GPQA may appear heterogeneous, its domain range is still relatively constrained (biology, physics, chemistry) compared to broader benchmarks. Following the reviewer’s suggestion, we additionally evaluate on ARC-Challenge [1], a multi-step commonsense reasoning benchmark with less explicit domain structure. We use the 299 samples from the official ARC validation set.
>
> | Method                        | ARC Acc. |
> |-------------------------------|----------|
> | Top-1 Model                    | 88.6     |
> | Self-Consistency (Top-1 ×5)    | 89.3     |
> | Mixture-of-Agents              | 90.1     |
> | ReConcile                      | 89.0     |
> | Symbolic-MoE                   | **91.7**     |
>
> These results show that, **even in a setting where the task is less heterogeneous, our method still achieves the best performance among the main baselines** (although the performance margin is smaller than what we observe on the more heterogeneous benchmarks in Table 1).
>
> &nbsp;
> &nbsp;
>
> ### **\[W4\] Dependence on the Validation Dataset**
>
> > if the validation set only contains math questions and hence only math-subject tags are generated, the model profiles would not be useful for other context.
>
> This concern can be mitigated by using a broad validation dataset. As shown in Table 2 (line 358-377) of the original submission, model profiles constructed from the diverse categories in MMLU-Pro generalize well to out-of-distribution tasks. Specifically, we build model profiles using only MMLU-Pro and AIME validation data, and then directly evaluate on OmniMATH. All multi-agent baselines similarly select their top-3 models using the same validation sets. Under this setup, **Symbolic-MoE is significantly more robust to domain shift than the baselines**–for instance, the strongest baseline (Debate) experiences a −14.81% accuracy drop relative to our method.
>
> While Symbolic-MoE does rely on a small validation set to construct the profiles, Table 2 (line 358-377) demonstrates that these profiles transfer effectively to unseen domains. Therefore, in cases where task-specific validation data is unavailable, one can simply begin with a broad validation dataset such as MMLU and still obtain strong generalization.
>
> &nbsp;
> &nbsp;
>
> ### **\[W5\] Comparison to Additional Baselines**
>
> We agree that including a routing baseline strengthens the evaluation. Therefore, we include a new routing baseline, **The Avengers \[1\]**, for comparison. We keep the model pool and validation set identical to those in Symbolic-MoE and swap in the clustering-based routing strategy used by The Avengers. The resulting performance is shown below.
>
> | Method        | MMLU-Pro | AIME  | GPQA  | MedMCQA | Avg.  |
> |---------------|----------|-------|-------|---------|-------|
> | The Avengers  | 60.64    | 46.67 | 52.31 | 53.09   | 53.18 |
> | Symbolic-MoE  | **63.71**    | **68.88** | **57.78** | **59.35**   | **62.43** |
>
> These results show that **our method substantially outperforms this newer baseline**. This may be because The Avengers require a larger validation set to learn reliable query embeddings and perform clustering (they mixed 15 datasets, totaling 11,837 samples), whereas we only need \~300 validation samples.
>
> Moreover, The Avengers compares against several routing baselines, including LLM Router, which prompts an LLM to route instances; RouterDC \[2\], which uses contrastive learning to learn embeddings for both instances and models; EmbedLLM \[3\], which applies a binary cross-entropy objective for routing; and MODEL-SAT \[4\], which converts candidate models’ capabilities into textual descriptions that are embedded and fed into a trainable LLM to predict model suitability for each query. Since The Avengers outperforms these baselines, and our method performs better than The Avengers, this suggests that our method is competitive with these baselines and transitively outperforms them as well.
>
> * \[1\] https://arxiv.org/abs/2505.19797
> * \[2\] https://arxiv.org/abs/2409.19886
> * \[3\] https://arxiv.org/abs/2410.02223
> * \[4\] https://arxiv.org/abs/2502.17282

---

> ### Author Response · Authors · 2025-11-21
> **Response to Reviewer GGut (Part 3)**
>
> > The authors could consider discussing and/or comparing with related work that single model, multi-instance LLM ensembles where diversity is based on the prompts to each instance.
>
> We thank the reviewer for highlighting these prompt-diversity ensemble methods. We briefly summarize each work and clarify how they differ fundamentally from Symbolic-MoE.
>
> PREFER \[3\] adopts an AdaBoost-style prompt ensemble, where prompts serve as weak learners refined through a feedback–reflect–refine loop using training errors and are combined via weighted voting. DIPPER \[1\] constructs a single-model ensemble by generating a large pool of prompts and selecting a diverse subset using the Fidelity-Adjusted Semantic Volume (FASV) criterion. Diversity of Thought \[2\] similarly induces diversity through prompting by eliciting different reasoning styles or personas from the same LLM.
>
> While valuable, these approaches operate entirely within the single-model, prompt-diversification setting, whereas Symbolic-MoE explores multi-model diversity, combining heterogeneous experts through instance-level skill routing. These methods ask whether we can prompt a single model in multiple diverse ways, whereas Symbolic-MoE asks whether we can leverage complementary expertise across heterogeneous models. This orthogonal perspective clarifies why prompt-based ensembles and our model-diversity framework address fundamentally different hypotheses.
>
> Regarding whether these approaches could serve as “Advanced Single-Model” baselines alongside Self-Refine or Self-Consistency (SC), several barriers prevent a fair integration:
>
> 1\. Reproducibility. DIPPER \[1\] and Diversity of Thought \[2\] do not release code, and key components (prompt generation, semantic embeddings, FASV hyperparameters, prompt-pool size) are underspecified, making reproduction impossible in the rebuttal period.
>
> 2\. Training vs. inference-only. PREFER \[3\] requires a training phase with instance reweighting and iterative prompt refinement using labeled data. In contrast, all our compared baselines (Self-Refine, Self-Consistency, MoA, ReConcile) are strictly inference-only methods that use the validation set purely for model or aggregator selection—not for learning, optimization, or prompt refinement. Incorporating PREFER would change the experimental paradigm.
>
> **We genuinely appreciate the reviewer bringing these important works to our attention and have added a discussion to the Appendix Q (lines 1067-1200, highlighted in blue)** that explicitly contrasts prompt-based ensemble methods \[1-3\] with our model-based approach, clarifying that they test orthogonal hypotheses (prompt engineering vs. multiple expert model collaboration) and acknowledging that both dimensions could potentially be combined in future work.
>
> * \[1\] https://arxiv.org/abs/2412.15238
> * \[2\] https://arxiv.org/abs/2310.07088
> * \[3\] https://arxiv.org/abs/2308.12033

---

> ### Author Response · Authors · 2025-11-26
>
> Dear Reviewer,
>
> Thank you again for your insightful comments and valuable suggestions.
>
> We submitted our detailed response to your comments a few days ago. To ensure we have sufficient time for a productive discussion before the deadline, we kindly want to check whether our response has sufficiently addressed your concerns. If any questions remain, we would be very happy to clarify them. If our responses have addressed your comments, we would be grateful if you could consider them in your final evaluation.

---

### Official Review · Reviewer_zGjf · 2025-10-31

**Soundness:** 3
**Presentation:** 3
**Contribution:** 3
**Rating:** 6
**Confidence:** 3

**Summary:**

SYMBOLIC-MoE is a test-time, skill-aware mixture-of-experts framework: a “keyword LLM” extracts discrete skills and, from a small validation set, builds per-model skill profiles. At inference, it extracts a sample’s skills, matches them to these profiles, probabilistically recruits a few experts by combining local skill fit with global competence, and uses a task-selected aggregator to synthesize their chain-of-thought outputs.

**Strengths:**

1.	**Clear problem formulation and intuitive approach.** Figures 1 and 2 vividly contrast “task-level fixed agents” with “instance-level, skill-based routing.”
2.	**Reporting with variability**: Results include standard deviations, which strengthen the empirical claims and improve interpretability and reproducibility.

**Weaknesses:**

1.	**Missing comparison to RL-based ensembles/routing.** The paper argues for SYMBOLIC-MoE mainly against all-LLM ensembles and multi-agent baselines, but does not compare to lightweight RL/bandit routing or RL-trained ensembling (e.g., [A] or [B]) that already offer strong cost–quality trade-offs.
2.	Aggregator not validated vs stronger fusion. Try the GenFuser [C] as the aggregator and provide an ablation to see if gains stem from fusion rather than routing.
3.	**Unclear online behavior with small/no batches.** In an online production setting with mixed, bursty traffic, how does your expert-grouped batching behave when batches are small or fail to form?


[A] Router-R1: Teaching LLMs Multi-Round Routing and Aggregation via Reinforcement Learning, NeurIPS 2025.

[B] Efficient Dynamic Ensembling for Multiple LLM Experts, IJCAI 2025.

[C] LLM-Blender: Ensembling Large Language Models with Pairwise Ranking and Generative Fusion, ACL2023.

**Questions:**

see Weaknesses.

---

> ### Author Response · Authors · 2025-11-21
> **Response to Reviewer zGjf (Part 1)**
>
> ### **\[W1\] Comparison to RL-based ensembles/routing**
>
> > compare to lightweight RL/bandit routing or RL-trained ensembling…
>
> We thank the reviewer for pointing out these relevant references. **We have included them in the revised PDF (highlighted in blue, lines 128-131).** We emphasize that our routing process differs substantially from RL-based ensemble/routing methods in both design and scope:
>
> 1) Our router is intentionally simple and efficient: it uses \~300 validation examples to build model profiles and performs categorical sampling. In contrast, training RL-based routers requires more data (in Router-R1, the authors mentioned *“we additionally construct a dedicated router training dataset. Specifically, each training question is independently fed to every model in the LLM routing pool multiple times with temperature sampling.”*) and access to online model feedback.
>
> 2) Router-R1 uses a substantially different model pool and trains a 3B LLM via RL, whereas our method is training-free. Their approach also relies on manually written LLM descriptions to characterize each model’s strengths, while our model profiles are derived automatically via annotated skills.
>
> 3) Router-R1 assumes access to online-serving commercial APIs (e.g., Together AI) for multiple LLMs in order to get online feedback. In contrast, we propose the batch inference strategy, which supports local deployment on a single GPU.
>
> We introduce a practical, training-free, instance-level routing framework for large collections of heterogeneous LLM experts, and a novel batching mechanism that makes this possible on a single GPU.
>
> While RL-based routing is somewhat orthogonal to our focus, we agree that including a routing baseline strengthens the evaluation. Therefore, we include a new routing baseline, **The Avengers \[1\]**, for comparison. We keep the model pool and validation set identical to those in Symbolic-MoE and swap in the clustering-based routing strategy used by The Avengers. The resulting performance is shown below.
>
> | Method        | MMLU-Pro | AIME  | GPQA  | MedMCQA | Avg.  |
> |---------------|----------|-------|-------|---------|-------|
> | The Avengers  | 60.64    | 46.67 | 52.31 | 53.09   | 53.18 |
> | Symbolic-MoE  | **63.71**    | **68.88** | **57.78** | **59.35**   | **62.43** |
>
> These results show that our method substantially outperforms this newer baseline. This may be because The Avengers require a larger validation set to learn reliable query embeddings and perform clustering (they mixed 15 datasets, totaling 11,837 samples), whereas we only need \~300 validation samples.
>
> Moreover, The Avengers compares against several routing baselines, including LLM Router, which prompts an LLM to route instances; RouterDC \[2\], which uses contrastive learning to learn embeddings for both instances and models; EmbedLLM \[3\], which applies a binary cross-entropy objective for routing; and MODEL-SAT \[4\], which converts candidate models’ capabilities into textual descriptions that are embedded and fed into a trainable LLM to predict model suitability for each query. Since The Avengers outperforms these baselines, and our method performs better than The Avengers, this suggests that our method is competitive with these baselines and transitively outperforms them as well.
>
> \[1\] [https://arxiv.org/abs/2505.19797](https://arxiv.org/abs/2505.19797)
> \[2\] https://arxiv.org/abs/2409.19886
> \[3\] https://arxiv.org/abs/2410.02223
> \[4\] https://arxiv.org/abs/2502.17282

---

> ### Author Response · Authors · 2025-11-21
> **Response to Reviewer zGjf (Part 2)**
>
> ### **\[W2\] Comparison of the Aggregator and Stronger Fusion**
>
> > Aggregator not validated vs stronger fusion… if gains stem from fusion rather than routing.
>
> We appreciate the reviewer’s suggestion to compare our aggregator against stronger fusion methods. In Table 4 (lines 394-408) of the original submission (reproduced below), we had a comparison among several aggregation strategies:
>
> | Expert           | Aggregator          | GPQA Acc. |
> |-----------------|-------------------|-----------|
> | Random           | Task-Specific      | 31.82     |
> | Recruited        | Random             | 51.52     |
> | Recruited        | Majority Vote      | 53.54     |
> | Recruited        | Task-Specific      | **57.78**     |
>
> When the aggregator is suboptimal, majority voting can serve as a robust alternative. However, when the expert models themselves are weak (chosen randomly), even a strong aggregator cannot compensate for the performance drop.
>
> In summary, both expert and aggregator selection are important. **Our recruited experts can already achieve improved performance with just Self-Consistency (53.54% vs. 51.52%), but can be further improved with a better chosen aggregator (57.78% vs. 53.54%).**
>
> > Try the GenFuser as the aggregator
>
> We attempted to include this comparison. However, the publicly released [GenFuser](https://github.com/yuchenlin/LLM-Blender) model is constrained to max 256 tokens for the input query and 256 tokens per candidate output, and it is trained primarily on instruction-following tasks but not reasoning tasks.  When applied to our setting, GenFuser yielded 0% accuracy, primarily because (1) the inputs are severely truncated, and (2) the domain mismatch prevents effective fusion.
>
> We therefore conclude that the released version of GenFuser is not directly suited for our reasoning-focused evaluation. We remain open to including it as a baseline in future work if a reasoning-oriented checkpoint with substantially larger context capacity becomes available.
>
> &nbsp;
> &nbsp;
>
> ### **\[W3\] Unclear online behavior with small/no batches.**
>
> In cases where expert-grouped batches do not form, e.g., when traffic is very sparse or mixed, we simply revert to the standard inference setup. Concretely:
> * With k GPUs, we load one expert model per GPU and run them independently.
> * With a single GPU, we load experts sequentially and collect their outputs.
>
> We would like to note that the expert-grouped batching strategy is an implementation-level optimization. That is, it is designed to improve efficiency under typical online traffic or benchmark evaluations. As with most batching methods, its efficiency benefits are less pronounced under extremely small or irregular batches, but it wouldn’t affect the correctness or operation of the system. **We thank the reviewer for raising this point, and have added this to the limitation section (lines 513-519, highlighted in blue).**

---

> > ### Comment · Reviewer_zGjf · 2025-11-25
> >
> > Thank you for your detailed responses. I have read through all other reviews and authors’ responses. After careful consideration, I will maintain my current rating.

---

> > > ### Author Response · Authors · 2025-11-26
> > >
> > > Dear Reviewer,
> > >
> > > Thank you for your positive assessment and insightful comments. Please feel free to reach out with any additional questions or suggestions.

---

### Official Review · Reviewer_DD14 · 2025-11-01

**Soundness:** 2
**Presentation:** 3
**Contribution:** 2
**Rating:** 2
**Confidence:** 3

**Summary:**

This paper proposes a model ensemble method that aggregates the outputs of multiple models to solve specific problems. The authors design a skill-based method to select appropriate LLMs for each instance. Specifically, every candidate LLM is assigned a profile containing scores for solving different kinds of tasks (skills), based on its performance on a validation set. When a new test instance is introduced, it is also labeled with relevant skills. The LLMs that have high scores on these specific skills are then selected to process the instance. Finally, another LLM is used to aggregate the outputs from the selected models and generate a final answer. Experiments reportedly show that this proposed ensemble method outperforms both single LLMs and previous ensemble methods.

**Strengths:**

This work addresses the critical routing problem within model ensemble or multi-agent scenarios. The proposed skill-based routing is interesting and novel, and it seems more appropriate than traditional task-level routing.

**Weaknesses:**

1. **Unfair Experimental Setting**: The experimental settings appear unfair. The authors include reasoning models in their candidate pool (e.g., QwenR1, LlamaR1) but compare them only against non-reasoning baselines (e.g., deepseek-v3, llama3-70B). This comparison is problematic, as reasoning models have a strong inherent advantage on tasks like AIME. As shown in Table 1, QwenR1 alone outperforms the non-reasoning baselines. Given this setup, the claim that the proposed method allows smaller LLMs to outperform models with more parameters is not well-supported. For a fair comparison, the authors should either include reasoning models in the baselines or remove them from the candidate pool.

2. **Lack of Clarity in Profile Creation**: The model profile creation stage is not clearly described. The authors state that an LLM is used to annotate the skills needed for each query in the validation set; however, the prompt used for this annotation is not provided. It is unclear whether the LLM's prediction is based on a predefined keyword pool. If no such pool is used, the LLM's outputs for semantically identical skills might be slightly different (e.g., "algebra" vs. "algebraic"). Do the authors employ a normalization method for these outputs? Furthermore, the distribution of skills in the validation set is missing. The model profile's reliability depends on having a sufficient number of test queries for every skill.

3. **Contradictory "Global Competency"**: The introduction of "global competency" seems contradictory to the principle of instance-based routing. If I understand correctly, global competency is not relevant to the current test instance; rather, it only reflects a model's general performance on the validation set. Why not rely solely on the local suitability score?

4. **Expert Sampling Method**: The paper states that K experts are sampled **with replacement**, which means a single expert might be selected multiple times for one instance, effectively reducing the number of unique experts used. The softmax temperature is set to 0.5, which would make the expert selection probability distribution spiky. This setup would encourage the selection of only one or two experts with the highest probabilities, making the "mixture-of-experts" effectively a "select-the-best-expert" approach. Can the authors provide statistics on the average number of unique experts used per instance?

**Questions:**

Is the aggregator's selection ability also related to the skills required for each query? In lines 197-199, the authors state this is not a good choice, but they do not provide evidence to support this claim.

---

> ### Author Response · Authors · 2025-11-21
> **Response to Reviewer DD14 (Part 1)**
>
> We thank the reviewer for recognizing that our work “addresses the critical routing problem” and “is interesting and novel”. Please find the answers to your questions below.
>
> &nbsp;
> &nbsp;
>
> ###  **\[W1\] Clarification on the Experimental Setting**
>
> We thank the reviewer for the actionable suggestion and the opportunity to clarify this point for future readers. However, we believe the evaluation setting is fair and well-motivated for the following reasons:
>
> **(1) All the multi-agent baselines can use these reasoning models as well.**
>
> > For a fair comparison, the authors should either include reasoning models in the baselines…
>
> As shown in Table 1, **our baselines *do* incorporate reasoning-oriented models.** For example, the Self-Consistency baseline on AIME-2024 invokes QwenR1 five times and performs majority voting. Likewise, both MoA and ReConcile select their top-3 agents based on validation accuracy, and these sets include strong reasoning models such as QwenR1 and LlamaR1. Despite this, they still fall short of our method. Therefore, the baseline comparison is already fair and appropriately controlled.
>
> **(2) QwenR1 is not universally strong and, by itself, cannot compete with frontier or larger models.**
>
> While QwenR1 7B performs well on AIME, it significantly lags behind models such as Gemini 1.5 Pro, DeepSeek V3, Qwen2.5-70B, and Llama3.3-70B on all other benchmarks (MMLU-Pro, GPQA, and MedMCQA). As shown in Table 1 (line 270, last column), QwenR1 7B achieves **48.04%**, whereas Gemini 1.5 Pro and DeepSeek V3 reach **61.84%** and **59.12%**, and larger open-source models achieve **53–54%**. Thus, simply including QwenR1 7B in the candidate pool does *not* create an inherently unfair advantage, i.e., QwenR1 7B alone is not competitive with these frontier baselines across the full suite of tasks.
>
> **(3) The distribution of the recruited experts confirms that reasoning models do not dominate the pool.**
>
> Figure 4 shows the expert-selection distribution. Importantly, for benchmarks such as MMLU-Pro and MedMCQA, **neither QwenR1 nor LlamaR1 is selected**, demonstrating that the symbolic routing naturally avoids these reasoning models when they are not advantageous. This indicates that our gains are not driven by always picking reasoning-heavy models, and they are not always dominant in the model pool.
>
> &nbsp;
> &nbsp;
>
> ### **\[W2\] Clarification on the Profile Creation**
>
> > However, the prompt used for this annotation is not provided.
>
> We clarify that we have provided all the prompts we used in Appendix O (line 1080\) in the original submission.
>
> > It is unclear whether the LLM's prediction is based on a predefined keyword pool. If no such pool is used, the LLM's outputs for semantically identical skills might be slightly different (e.g., "algebra" vs. "algebraic").
>
> As described in lines 180-186 of the original submission, our annotation process does not rely on any predefined keyword pool, and we do not manually normalize keyword variants. Instead, for each sample, **we query the LLM five times and retain only the keywords that appear at least once**, which effectively filters out inconsistent or overly specific variants.
>
> > Furthermore, the distribution of skills in the validation set is missing.
>
> To further address concerns regarding the reliability of model profiles, we provide below a preview of the skill distributions in the validation sets. These show that the annotated skills are diverse but well represented.
>
> &nbsp;
> &nbsp;
>
> **AIME 2024 (Unique keywords: 75)**
>
> | Skill         | Count | Percent |
> |---------------|-------|---------|
> | algebra       | 15    | 9.43%   |
> | number theory | 13    | 8.18%   |
> | geometry      | 12    | 7.55%   |
> | combinatorics | 9     | 5.66%   |
> | trigonometry  | 6     | 3.77%   |
>
> &nbsp;
>
> **MMLU-Pro (Unique keywords: 1053)**
>
> | Skill          | Count | Percent |
> |----------------|-------|---------|
> | psychology     | 185   | 1.66%   |
> | finance        | 184   | 1.65%   |
> | economics      | 159   | 1.42%   |
> | thermodynamics | 134   | 1.20%   |
> | law            | 112   | 1.00%   |
>
> For the expanded distribution on all benchmarks, we have included them in the revised PDF (Appendix K and Figure 6).

---

> ### Author Response · Authors · 2025-11-21
> **Response to Reviewer DD14 (Part 2)**
>
> ### **\[W3\] Rationale behind "Global Competency"**
>
> > If I understand correctly, global competency is not relevant to the current test instance; rather, it only reflects a model's general performance on the validation set. Why not rely solely on the local suitability score?
>
> Your understanding of *global competency* is correct: it is not instance-dependent, but instead reflects a model’s overall capability estimated from a validation set. In expert routing, there is an inherent trade-off between selecting models with **highly specialized strengths** (captured by the local suitability score) and selecting models that are **consistently strong** (captured by global competency). Relying solely on local suitability can lead the router to prefer models that excel at niche patterns but underperform more broadly, whereas incorporating global competency helps maintain a balanced selection.
>
> To validate this design choice, we conduct an ablation comparing Symbolic-MoE with and without global competency:
>
> |                               | AIME  | MMLU-Pro | MedMCQA | GPQA  | Avg   |
> |-------------------------------|-------|----------|---------|-------|-------|
> | w/o Global Competency         | **68.88** | 61.32    | **60.11**   | 56.32 | 61.66 |
> | w/ Global Competency (ours)  | **68.88** | **63.71**    | 59.35   | **57.78** | **62.43** |
>
> While routing based only on local suitability is indeed feasible, as shown above, in the majority of cases (except for MedMCQA), as well as in the averaged performance, taking global competency into account yields a slight improvement across the four datasets. This supports our argument that balancing instance-level specialization with overall capability leads to more robust routing behavior.
>
> ### **\[W4\] Clarification on the Expert Sampling Method**
>
> > The paper states that K experts are sampled with replacement, which means a single expert might be selected multiple times for one instance, effectively reducing the number of unique experts used.
>
> It is correct that we sample *k* experts with replacement. **However, it is not always selecting the same model.** This choice is intentional and is supported by our ablation results in Table 5 (line 410\) in our original submission (reproduced below for convenience):
>
> | Recruiting Strategy  | Accuracy (%) |
> |----------------------|--------------|
> | Top-3 Experts        | 52.86        |
> | Top-5 Experts        | 47.68        |
> | 3 Random Experts     | 42.61        |
> | 5 Random Experts     | 44.92        |
> | Model Profile (Ours) | **57.78**        |
>
> Forcing *k* distinct experts can be suboptimal: in many cases, the top 1–2 experts are significantly more competent than the remaining pool. **Sampling with replacement naturally accommodates this scenario by allowing multiple draws of the same strong experts when they dominate an instance.** At the same time, when the expert distribution is flatter (i.e., several experts perform similarly), the sampling process still yields a diverse set of experts.
>
> We agree that this reduces the number of unique experts per instance. However, this behavior is *intended* and reflects the underlying structure of the expert performance distribution. Our ablations demonstrate that this strategy consistently outperforms enforcing unique experts.
>
> In response to the reviewer’s request, we provide additional statistics on the average number of unique experts per sample (i.e., \# of unique experts / \# of samples) for four benchmarks as below.
>
> | Dataset   | Avg. Num. of Unique Experts |
> |-----------|------------------------------|
> | AIME      | 1.67                         |
> | MMLU-Pro  | 2.14                         |
> | MedMCQA   | 2.35                         |
> | GPQA      | 1.62                         |
>
> When recruiting k=3 experts with replacement, it is not always selecting the same model and thus, on all benchmarks, the average number of unique experts \> 1.5. Moreover, on more heterogeneous tasks such as MMLU-Pro and MedMCQA, there are more than two distinct experts being selected, even though we allow replacement.

---

> ### Author Response · Authors · 2025-11-21
> **Response to Reviewer DD14 (Part 3)**
>
> ### **\[Q1\] Is the aggregator’s selection ability also related to the skills required for each query?**
>
> The short answer is: no. **Aggregator selection depends on how well a model can consolidate *k* expert outputs into a final correct answer.** Our results suggest that aggregation ability is largely independent of reasoning ability. For example, the robustness of Majority Vote indicates that aggregation ≠ domain expertise (Table 4). **In Table 3, we also tried selecting the aggregator using the same skill-based strategy as expert selection (“adaptive’’ in Table 3), but this approach performs worse in practice.**
>
> Intuitively, effective aggregation requires interpreting heterogeneous CoTs, identifying inconsistencies, and resolving conflicting rationales—capabilities that do not benefit from aligning the aggregator’s skills with those of the experts. This explains why the skill-based adaptive aggregator underperforms (c.f Table 3\) and why Majority Vote remains competitive (c.f Table 4\) despite being completely skill-agnostic.
>
> Together, these findings show that good reasoners are not necessarily good aggregators, supporting our task-level aggregation strategy. We hope this helps clarify the point!

---

> ### Author Response · Authors · 2025-11-26
>
> Dear Reviewer,
>
> Thank you again for your insightful comments and valuable suggestions.
>
> We submitted our detailed response to your comments a few days ago. To ensure we have sufficient time for a productive discussion before the deadline, we kindly want to check whether our response has sufficiently addressed your concerns. If any questions remain, we would be very happy to clarify them. If our responses have addressed your comments, we would be grateful if you could consider them in your final evaluation.

---

### Author Response · Authors · 2025-11-26
**General Response**

We appreciate the time and effort in reviewing our manuscript. As highlighted by reviewers, Symbolic-MoE provides a clear and intuitive skill-aware mixture-of-experts framework that effectively addresses the critical routing problem in multi-model and multi-agent settings (**DD14, GGut, mHRt**). Our paper clearly formulates the need for instance-level, skill-based routing beyond traditional task-level methods (**DD14, zGjf**) and proposes an intuitive, reproducible pipeline (**GGut, mHRt**) with clear presentations and motivating examples (**zGjf, mHRt**). Under extensive experimental evaluation across diverse reasoning domains (math, science, medical), Symbolic-MoE delivers strong and consistent empirical improvements over single-model baselines and existing multi-agent approaches (**DD14, GGut, mHRt**), with supportive ablations and signs of cross-domain generalization (**mHRt**).

In response to the reviewers’ comments, we have revised and strengthened the manuscript with the following updates:

* **\[Reviewer zGjf W1\]:** Included related works pointed out by the reviewers (Section 2\)
* **\[Reviewer zGjf W3\]:** Added a discussion on the batch inference strategy
* **\[Reviewer DD14 W2\]:** Added keyword distribution of the validation data (Appendix K)
* **\[Reviewer GGut W5\]:** Added discussion on diversity-based prompt example (Appendix Q)

These revisions are temporarily highlighted in blue for easier reference. Beyond the manuscript updates, we additionally conducted the following experiments during the rebuttal period:

* **\[Reviewer mHRt Q1 & Reviewer DD14 W1\]:** Ablations on the Global Competency
* **\[Reviewer mHRt W4 & Reviewer zGjf W1\]:** Ablations on the routing strategies
* **\[Reviewer DD14 W4\]:** Ablations on the expert sampling method
* **\[Reviewer GGut W3\]:** Sensitivity analysis on the quality of the skill tags
* **\[Reviewer zGjf W1 & Reviewer GGut W5\]:** Comparison to an external routing baseline
* **\[Reviewer mHRt W3 & Reviewer GGut W2 & Reviewer zGjf W2\]:** Ablations on different aggregation strategies
* **\[Reviewer GGut W3\]:** Additional experiment on a non-heterogeneous, non-math task
* **\[Reviewer mHRt W2\]:** Analysis of profile reliability versus validation size
* **\[Reviewer mHRt W5\]:** Operational cost report
* **\[Reviewer mHRt W6\]:** Qualitative error analysis
* **\[Reviewer mHRt W6\]:** Robustness check with alternative embedders (E5/SimCSE)

We believe that these changes address the comments made by reviewers. We thank the reviewers again for their feedback, which has helped us further strengthen the paper.

Authors

---

### Meta-Review · Area_Chair_UEpm · 2026-01-10

**Summary:**

The paper's idea is quite simple and straightforward. It tries to learn a symbolic way to assign tasks to agents of different expertises. Despite simplicity, it raises a few additional questions:

- It inevitably loses the generality of different agents and the pipeline is mostly suitable for some very specialize tasks.

- The granularity of an agent's expertise will be quite subjective and hard to specify in practice.

- The capability will be limited by the router. Once the router is insufficiently capable, then the multi-agent framework could perform poorly

After carefully reading the paper and review, I agree with some reviewers' concerns, such as (1) lack of baselines (it mostly compares with MoE), which is not sufficient; (2) the framework is a heuristic design rather than a systematic and principled one. Combining pieces, I find this paper, at its current form, below the bar of acceptance.

**Reviewer Concerns:**

The authors did a decent job in participating in the rebuttal stage, but I think the empirical validation can be performed in a more comprehensive way. Thus it will be better for the paper to undergo another round of review.

**Reviewer Scores:**

The final score is 6,4,4,2. Three negative reviewers didn't raise their scores, and the positive reviewer also didn't champion the paper.

---

### Decision · Program_Chairs · 2026-01-26

Reject